



# Spatially and temporally resolved measurements of NOx fluxes by airborne eddy-covariance over Greater London

Adam R. Vaughan[1], James D. Lee[1,2], Stefan Metzger[3,4], David Durden[3], Alastair C. Lewis[1,2], Marvin D. Shaw[1,2], Will S. Drysdale[1,2], Ruth M. Purvis[1,2], Brian Davison[5] and C. Nicholas Hewitt[5]

[1]Wolfson Atmospheric Chemistry Laboratories, Department of Chemistry, University of York, York, YO10 5DD, UK
[2]National Centre for Atmospheric Science, University of York, York, YO10 5DD, UK
[3]National Ecological Observatory Network Program, Battelle, 1685 38th Street, Boulder, CO 80301, USA
[4]Department of Atmospheric and Oceanic Sciences, University of Wisconsin-Madison, 1225 West Dayton Street, Madison,
WI 53706, USA
[5]Lancaster Environment Centre, Lancaster University, Lancaster, UK

*Correspondence to*: Adam R. Vaughan (adam.vaughan@york.ac.uk)

**Abstract.**

Flux measurements of nitrogen oxides (NOx) were made over London using airborne eddy-covariance from a low flying aircraft. Seven low altitude flights were conducted over Greater London performing multiple over-passes across the city during eight days in July 2014. NOx fluxes across the Greater London region exhibited high heterogeneity and strong diurnal variability, with central areas responsible for the highest emission rates (20 - 30 mg m$^{-2}$ h$^{-1}$). Other high emission areas included the M25 orbital motorway. The complexity of London's emission characteristics makes it challenging to pinpoint single 20 emission sources definitively using airborne measurements. Multiple sources, including road transport and residential, commercial and industrial combustion sources are all likely to contribute to measured fluxes. Measured flux estimates were compared to scaled National Atmospheric Emissions Inventory (NAEI) estimates, accounting for; monthly, daily and hourly variability. Significant differences were found between the flux-driven emissions and the NAEI estimates across Greater London, with measured values up to two times higher in Central London than those predicted by the inventory. To overcome 25 the limitations of using the national inventory to contextualise measured fluxes, we used physics-guided flux data fusion to train environmental response functions (ERF) between measured flux and environmental drivers (meteorological and surface). The aim was to generate time-of-day emission surfaces using calculated ERF relationships for the entire Greater London region (GLR). 98% spatial coverage was achieved across GLR at 400 m$^2$ spatial resolution. All flight leg projections showed substantial heterogeneity across the domain, with high emissions emanating from Central London and major road 30 infrastructure. The diurnal emission structure of the GLR was also investigated, through ERF, with the morning rush-hour distinguished from lower emissions during the early afternoon. Overall, the integration of airborne fluxes with an ERF-driven strategy enabled the first independent generation of surface NOx emissions, at high resolution using an eddy-covariance approach, for an entire city region.



## 1 Introduction

Anthropogenic emissions of $NO_x$ (NO + $NO_2$ = $NO_x$) occur over large areas of Europe and the United Kingdom, with atmospheric concentrations in many urban areas exceeding the recommended World Health Organisation (WHO) 40 µg m$^{-3}$ annual health limit value (Brookes et al., 2013). Of all the common gaseous air pollutants, nitrogen dioxide ($NO_2$) is particularly problematic as it promotes respiratory diseases, such as lung inflammation, bronchial reactivity and a significant reduction in lung capacity (Foster et al., 2000; Kelly and Fussell, 2017; Shao et al., 2019). $NO_2$ also plays a central role in the production

of ground-level ozone at the regional scale. London has operated a low emission zone (LEZ) since 2008, with the aim of reducing air pollution through vehicle-specific restrictions. The effectiveness of the current LEZ on respiratory health is still unclear, with some studies highlighting the need further to reduce $NO_2$ concentrations, before improvements in public health are achieved (Mudway et al., 2019). Analysis of UK and European road-side $NO_x$ annual trends have shown a downward trend in $NO_2$ concentrations, however; road-side concentrations in regions such as Greater London remain well above WHO

guidelines as of 2020 (Grange et al., 2017; Lang et al., 2019).

In order to bring atmospheric concentrations of air pollutants into alignment with air quality standards, it is first necessary to understand where the pollutant originates from so that effective legislative controls can be introduced. The National Atmospheric Emissions Inventory (NAEI) is the primary tool used by the UK Government for this purpose. A growing body

of work has been conducted to evaluate the NAEI, by comparing inventory estimates with real-time flux measurements from towers and airborne platforms (Björkegren and Grimmond, 2018; Famulari et al., 2010; Font et al., 2015; Langford et al., 2009, 2010; Lee et al., 2015; Pitt et al., 2019; Vaughan et al., 2016, 2017).

Inventory validation is a vital component towards reducing urban pollutant concentrations, requiring a continued understanding

of significant emission sources and spatial distributions. Eddy-covariance (EC) is a well-documented technique for quantifying atmospheric emission rates within the atmospheric boundary layer (Aubinet et al., 2012). Initially, EC studies focused on greenhouse gas emission assessment (Baldocchi, 2003), but these have now been extended to include reactive atmospheric compounds such as volatile organic carbon compounds (VOCs) and $NO_x$ (Baldocchi, 2003; Karl et al., 2001, 2017, 2002; Langford et al., 2009, 2010; Lee et al., 2015; Marr et al., 2013; Squires et al., 2020; Vaughan et al., 2016). Here we present a

new methodology for calculating high spatial resolution $NO_x$ fluxes by airborne eddy-covariance and use these with other techniques to generate real-time emission grids over complex urban terrain. The method is demonstrated for the Greater London region but will be applicable to other metropolitan areas worldwide.





## 2 Methods

### 2.1 Measurement campaign

Airborne eddy-covariance measurements were made during seven research flights as part of the Ozone Precurers Fluxes in an Urban Environment (OPFUE) project in July 2014 (Shaw et al., 2015; Vaughan et al., 2016, 2017). The project involved multiple low altitude flights over the Greater London Region (GLR) using the Natural Environment Research Council's (NERC) Dornier-228 aircraft, based at Gloucestershire Airport's Airborne Research and Survey Facility (ARSF). The aircraft has a maximum flight range of 2,600 km and science ceiling altitude of 4,500 m.


Each research flight consisted of the following structure. An initial profile to 2,600 m was carried out at the beginning of each flight, allowing for calibrations in lower-$NO_x$ air during the transit towards London. After transitting, a spiral descent over Goodwood (SE England), gave an estimation of boundary layer height. Straight level transects at 300-400 m were then flown across Greater London, starting at the southwest corner of the M25 orbital motorway and finishing at the opposite northeast

edge of the GLR. A sharp right turn was then made towards the industrial areas of east London and over the Dartford Thames river crossing. The final transect ran perpendicular to the original, ending at the northwest corner of London, completing an open figure-of-eight design. The loop was not completed around the West of London, due to Heathrow airport. Each flight contained three repeat passes. Fig. 1 shows the flight path, with each transect type labelled. Table 1 summaries each transect type, the typical flight distance, location and the number of completed replicates.


| transects | length/ m | start | finish | area type | replicates |
|:---:|:---:|:---:|:---:|:---:|:---:|
| 1 | 50 km | 51.30° N, 0.45° W | 51.60° N, 0.18° E | suburban & urban | 14 |
| 2 | 30 km | 51.40° N, 0.20° E | 51.62° N, 0.25° E | suburban & urban | 5 |
| 3 | 30 km | 51.40° N, 0.20° E | 51.65° N, 0.15° E | suburban & urban | 10 |
| 4 | 13 km | 51.60° N, 0.10° E | 51.50° N, 0.30° E | urban (major roads) | 13 |
| 5 | 14 km | 51.50° N, 0.30° E | 51.40° N, 0.20° E | urban (major roads & industry) | 16 |

**Table 1. Transect information**

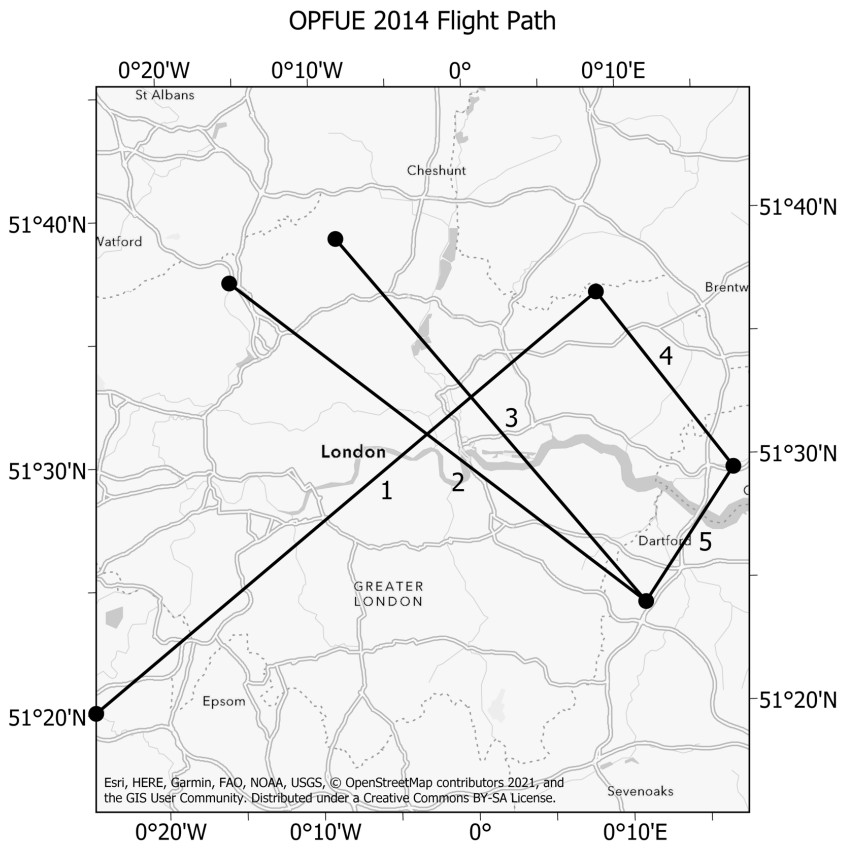

**Figure 1. OPFUE 2014 flight path over Greater London, highlighting the incomplete figure of eight structure. Each transect type has been labelled. Plotted in ArcGIS® (Esri, 2021).**

## 2.2 Instrumentation

Eddy-covariance flux measurements of $NO_x$ were made using an Air Quality Design Inc. (Golden, Colorado, USA) $NO_x$ chemiluminescence analyser (Fast-AQD-$NO_x$). The instrument has a dual-channel architecture capable of quantifying ambient mixing ratios of NO and $NO_2$ sequentially at 9 Hz (Squires et al., 2020). Fig. 2 depicts the flow schematic for the instrument, showing two separate detection channels for NO and $NO_2$. Sample inlet pressure is kept at 266 hPa to negate the effect of changing altitude on instrument sensitivity. NO is quantified by the ozone-chemiluminescence reaction, with 100 sccm of $O_3$ added to a 1500 sccm sample flow (Drummond et al., 1985; Kley and McFarland, 1980; Lee et al., 2009; Reed et al., 2016). Quantification of $NO_2$ mixing ratios follows an identical pathway, with an added conversion step. Ambient $NO_2$ is first photolytically converted to NO using a blue-light converter. The converter consists of a Teflon block containing a milled cavity (10 mL volume) down its centre, and three $395 \pm 20$ nm wavelength LED positioned on either side (Reed et al., 2016). Below 400 nm $NO_2$ photolytically degrades to NO and ground-state molecular oxygen (O[$^3$P]). The converter is cooled using a Peltier





cooler to reduces thermal interference products and exhibits a conversion efficiency of >85 % and residence time of 0.11 s. After conversion, detection is achieved using the same ozone-chemiluminescence reaction as NO. Chemiluminescence detection is achieved using dry-ice cooled (-60 °C) photomultiplier tubes (PMTs) (Hamamatsu Photonics K. K.) with a red-window filter.


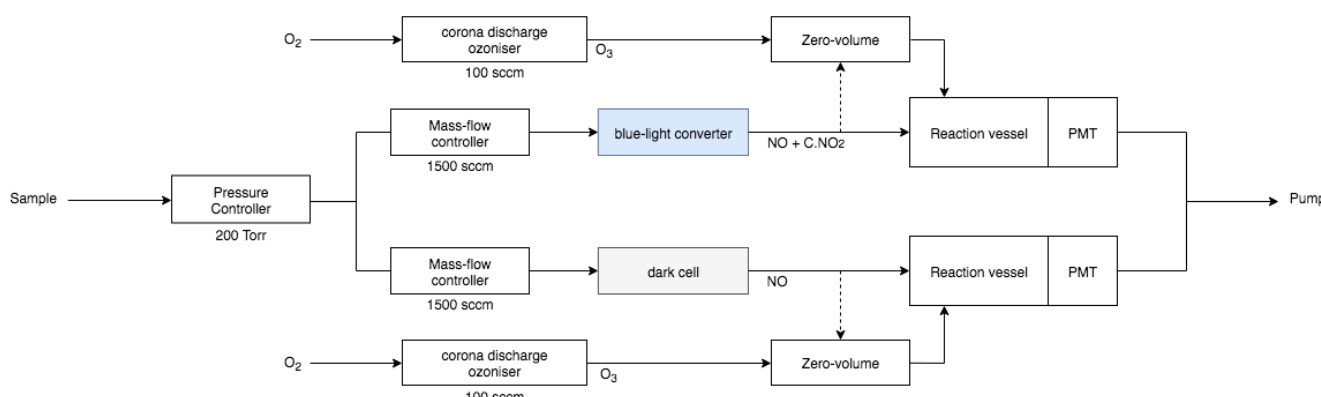

**Figure 2. Instrument schematic for fast AQD dual-channel chemiluminescence NO$_x$ analyser (Fast-AQD-NO$_x$). Dotted flow path represents zero count-rate flow path for both channels, giving a zero-count rate for each PMT.**

To assess the dark count rate on each PMT, the zero-volume flow path is used. Mixing between the sample flow and O$_3$ occurs in the zero-volume, ensuring the chemiluminescence reaction occurs before the sample reaches the reaction vessel. Typical PMT dark counts ranged from 2,000-3,000 counts s$^{-1}$. The dark-count rate on each PMT was measured frequently and subtracted from the ambient signal to give an accurate background correction. Instrument design takes into account water vapour quenching with regards to photon counting (Ridley et al., 1992). A constant amount of deionised water vapour (d.H$_2$O)

is added to the O$_3$ supply, increasing the concentration within the reaction vessel to ~32 parts per thousand (ppth). By standardising water vapour within the reaction vessel, any changes in atmospheric water vapour concentration are negated. Using Eq. (1), a corrected dry mixing ratio can be calculated from the wet mixing ratio and percentage of water vapour in the reaction vessel. On average, a 3.3 % correction was required. As this correction is small compared to the measurement uncertainties, it was not applied.


$$dry_{mixing\ ratio} = \frac{wet_{mixing\ ratio}}{1-0.1(H_2O\%)},$$ (1)

The sensitives of both the PMTs and the blue-light converter can drift over time, requiring calibration to a known NO/NO$_2$ standard. A 5 ppm NO standard (BOC Group plc., supplied and certified) was used to calibrate against, which was further

certified against a high accuracy National Physical Laboratory (NPL) standard, before and after the field campaign. Instrument mass-flow controllers were calibrated before and after field campaigns using a gilibrator (high accuracy electric bubble flow



meter). We determine PMT sensitivity by standard addition of a small flow (5 - 10 sccm) of NO calibrant gas to a flow of $NO_x$-free air. $NO_x$ free air was obtained either by flying above the boundary layer where $NO_x$ levels are low or by removal using a Sofnofil/charcoal trap attached to the sample inlet. This gave a NO mixing ratio in the range 5 – 10 ppt. NO channel

sensitivity values range from 7 - 8 counts ppt$^{-1}$. $NO_2$ detector sensitivity was also determined by the same method, with typical value ranges from 9 - 11 counts ppt$^{-1}$. In addition to detector sensitivity, the conversion efficiency of the blue-light converter was also assessed. A known $NO_2$ mixing ratio was generated by titration reaction of NO calibrant gas with $O_3$ (5 sccm flow of $O_2$ by a UV pen ray lamp). The converter was found to give > 85% conversion efficiency during the entire campaign.

Instrument precision was quantified by assessing the dark count noise on each PMT, through sampling zero-air (Lee et al., 2009). Photon counting is a well-established technique, with rates following a Poisson distribution (Ingle and Crouch, 1972; Williamson et al., 1988). Allan variance analysis (Werle, 2011) of 1 hour of 'zero-air' data, gave a $2\sigma$ precision (limit of detection) for NO and $NO_2$ of 49 and 78 pptv, at an integration period of 9 Hz. By time averaging to 10 – 20 s, $2\sigma$ LODs improved significantly to below 10 ppt. A secondary precision analysis was conducted by Gaussian distribution analysis. A

Gaussian distribution was used over a Poisson as the count rate (>3,000 counts s$^{-1}$) was high enough to ensure both distributions become identical (Lee et al., 2009; Silvia and Skilling, 2006). The 9 Hz $2\sigma$ LODs for NO and $NO_2$, were found to be 49 and 79 pptv. Both precision approaches give similar LOD values and highlight the sensitivity of the instrument during fast data acquisition.

Instrument accuracy was assessed for systematic uncertainties. Sources of instrument inaccuracy were mass-flow controllers, calibration standards, the blue-light converter and channel artefacts. Instrument mass-flow controllers are accurate to ± 1% (manufacturer quoted). The $NO/N_2$ calibration standard has a quoted accuracy of ± 1% (supplied by BOC Group plc). The blue-light converter gives consistent, stable calibrations with an accuracy of ± 10% derived from signal stability of the CE calculation. By taking the root sum of the squares of all accuracy and precision errors, overall uncertainty was calculated. Total

uncertainty for a 100 ppt measurement of NO and $NO_2$ is 54.5% and 88.3% (at 9 Hz acquisition rate). For 1 ppb, the total uncertainty for NO and $NO_2$ is 5.9% and 13.5% (at 9 Hz acquisition rate).

In addition to the Fast-AQD-$NO_x$, on-board instrumentation also included a Proton-Transfer-Reaction Mass-Spectrometer (PTR-MS; Ionicon GmbH), a PICARRO greenhouse gas analyser, an Inertial-Position and Altitude System (IPAS 20) and an

Aventech Research Inc. Aircraft Integrated Meteorological Measurement System (AIMMS-20). The AIMMS-20 system delivers 20 Hz measurements of u,v,w wind vectors, temperature, pressure and relative humidity. The probe consists of five pitot-static pressure ports, configured in a cruciform array, giving horizontal and vertical wind speed measurements. The temperature and humidity sensors are located at the back of the probe in a reverse-flow housing to reduce particulate contamination (Beswick et al., 2008). The probe was calibrated for static and dynamic upwash (Vaughan et al., 2016, 2017).




## 2.2 Eddy covariance with Environmental Response Functions

Environmental Response Function (ERF) is a physics-guided flux data fusion designed to create a bridge from EC measurements to model grid-scale flux estimates (Metzger, 2018; Metzger et al., 2013; Xu et al., 2017, 2018). In ERF, high-rate time-frequency wavelet decomposition and flux footprint modelling are used to create a time-aligned dataset between response (flux) and driver (e.g., concentration, building height etc.) observations. From this time-aligned dataset, machine learning extracts a driver-response process model – outputting a multi-dimensional surface that connects flux to process. ERF then uses this driver-response process model to project flux maps with hourly and sub-kilometre resolution, extending the areal representation of the airborne $NO_x$ fluxes from few square kilometres around the flight tracks to the GLR. The following subsections detail the software used for ERF EC data processing and the principal processing steps.

### 2.2.1 eddy4R eddy-covariance software

Flux processing was achieved using the eddy4R software. The eddy4R family of R packages (Core Team, 2019) creates a modular function-based software solution for EC data processing, as described in Metzger et al., (2017). A development and systems operation approach (DevOps) was utilised to create reproducible, open-source, and extensible software that is version controlled. This DevOps schema enables a release and iteration cycle that, to date, has yielded the eddy4R.base, eddy4R.qaqc, and eddy4R.stor packages on a publicly available GitHub repository (Metzger et al., 2019; Xu et al., 2019). This modular framework facilitates scientific community-driven code development that extends the eddy4R software suite's capabilities. In the present study, we extended eddy4R to handle fluxes from a wide variety of chemical species by adding to the eddy4R.turb package that is currently in development.

To ensure portability and reproducibility, the eddy4R packages integrate into a Docker image, which builds upon a Linux computational environment and resolves all system and R dependencies (https://www.docker.com/why-docker). The Docker image hardens the code against operating system-induced anomalies and streamlines the code base to the essential requirements for processing. GitHub automatically triggers, and version controls Docker image builds, which are housed on Dockerhub. Continuous integration testing through Travis-CI and subsequent code reviews complete the build chain. This fosters rapid code development across teams and functionalities while mitigating unintentional errors that could corrupt the main codebase. The eddy4R-Docker DevOps framework thus provides a foundation for the distributed development of novel algorithmic solutions and their scalable execution. The described approach provides the end-user with a practical approach towards version control and result reproducibility.



### 2.2.2 Wavelet time-frequency decomposition

The $NO_x$ flux calculation was based on the wavelet eddy covariance (EC) approach discussed by Metzger et al. (2013) and has been described in detail elsewhere (Karl et al., 2013; Misztal et al., 2014; Thomas and Foken, 2007; Torrence and Compo, 1998; Wolfe et al., 2015; Yuan et al., 2015).

Wavelet EC uses continuous wavelet transformation (CWT) to extract time-frequency or space-wavenumber information from
atmospheric signals. For this study, the Morlet wavelet (Cohen, 2019) was chosen due to its strong track record in quantifying atmospheric turbulence (Karl et al., 2013; Thomas and Foken, 2005). The complete covariance between two signals ($x$ & $y$) is deduced by examining global covariance across all frequency scales, as shown in Eq. (2). $a_j$ defines the frequency domain scales, $b_n$ the time-domain scales, $\delta t$ the steps between time-domain scales, $\delta j$ the spacing between frequency domain scales, length of the data series ($N$) and $C_\delta$ wavelet specific reconstruction factor. CWT frequency scales are chosen so that the smallest
resolvable scale ($s_0$) is equal to $2\delta t$ (half sampling frequency) and the largest scale being $\delta j^{-1} \log_2(N\delta t/s_0)$. During CWT, the wavelet is scaled in both frequency and time domains, using a defined number of scales, Eq. (3-4). Frequency-domain scales increase exponentially from $j = 0$ to $J$ ($J$ being the Nyquist frequency). Time-domain scales are increased linearly, from $n = 0$ to $N$-1 ($N$ equal to the length of data series). A $\delta j$ value of 1/8 was chosen as a compromise between high-frequency resolution without long computational time. The average frequency-resolved coefficients over a selected segment of time give
a real covariance between two signals, which in turn is used to calculate the eddy−flux (Metzger et al., 2013). Summatively, this approach provides localised highly resolved fluxes whilst accounting for all relevant transport scales.

$$F = \overline{x'y'} = \frac{\delta j \delta t}{C_\delta N} \sum_{j=0}^{J} \sum_{n=0}^{N-1} \frac{w_x(a_j,b_n)w_y(a_j,b_n)^*}{a_j}, \tag{2}$$

$$a_j = a_0 2^{j\delta j}, \tag{3}$$

$$b_n = n\delta t, \tag{4}$$

### 2.2.3 Flux processing overview

Flux processing in eddy4R followed the workflow shown in Fig. 3. Individual transects were processed separately, with a
minimum flight distance of 15 km, ensuring large atmospheric transport scales were captured. Data periods containing sharp turns or orbital loops were omitted. Meteorology, position and concentration data were merged for each transect, giving a regularised data frame at 20 Hz. Each transect was screened for data outside of defined thresholds and omitted. Overall data pass rate was set to $\geq 90$ %. Successful transects underwent de-spiking using the method outlined by Brock (1986) in the form of Starkenburg et al. (2016) for wind vectors (u,v,w), temperature and NO & $NO_2$ mixing ratios. The technique is sensitive to





up to 4 consecutive data spikes. High-pass filtered cross-covariance maximisation (Hartmann et al., 2018) was applied to correct NO/NO$_2$ mixing ratios and air temperature for differences in sampling time compared to the vertical wind (w). Once lag-time corrected, data were resampled from 20 Hz to 9 Hz using mean rolling averaging (Zeileis and Grothendieck, 2005).

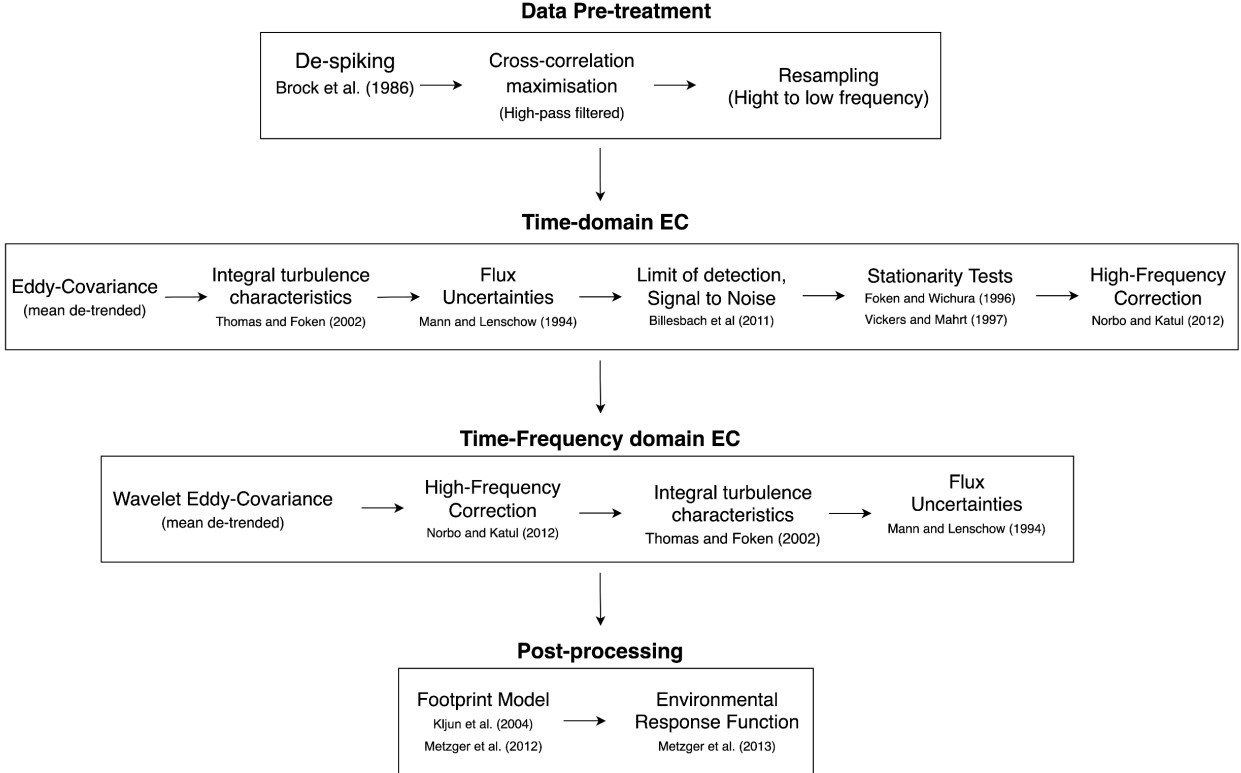

**Figure 3. Modular eddy4R workflow giving four processing steps: raw data pre-treatment, time-domain EC, time-frequency-domain EC, and post-processing analysis (footprint and ERF).**

After data pre-treatment, time-domain (classical) and time-frequency domain (wavelet) fluxes were calculated as outlined in Fig. 3. Time-domain EC gives a single flux estimate per transect, whereas time-frequency EC gives a flux measurement every

400 m along the transect using a 4000 m moving window. Time-frequency EC using CWT for flux analyses. A wavelet $\delta j$ value of 1/8 and a minimum wavelet scale of 4.5 Hz (Nyquist frequency) was chosen for wavelet calculations. Wavelet cone of influence was not removed in accordance with Metzger et al., (2013). Table 2 outlines eddy4R processing parameters.






| eddy4R parameter | Setting |
|---|---|
| *Data Frequency* | 9 Hz |
| *Transect length* | > 15 km |
| *De-spiking* | Median filter (Brock, 1986; Starkenburg et al., 2016) |
| *Lag correction* | High-pass filtered cross-correlation maximisation (Hartmann et al., 2018) |
| *De-trending* | mean |
| *High-Frequency Correction* | Yes (Nordbo and Katul, 2013) |
| *Wavelet waveform* | Morlet |
| *Wavelet $\delta j$* | 1/8 |
| *Wavelet maximum scale* | 512 s |
| *Wavelet COI inclusion* | yes |
| *Flux subinterval window* | 4,000 m |
| *Flux spatial averaging* | 400 m |

**Table 1.** List of eddy-covariance parameters for quantifying airborne $NO_x$ fluxes.


Each flight leg underwent the following QA/QC steps. Limit of detection (LOD) (Billesbach, 2011) and signal to noise (S/N) statistics (Foken and Wichura, 1996; Vickers and Mahrt, 1997) were calculated and median flux LODs were found to be 0.19 mg m$^{-2}$ h$^{-1}$ for NO and 0.57 mg m$^{-2}$ h$^{-1}$ for $NO_2$. Fluxes below these thresholds were flagged. Median S/N statistics for NO and $NO_2$ fluxes were found to be 14.54 and 17.26. Stationarity tests were calculated for each flight transect, with a flag threshold of 100% used (Foken and Wichura, 1996; Vickers and Mahrt, 1997). Nine out of 42 transects failed the stationarity criteria and so were omitted. NO and $NO_2$ fluxes were assessed for high-frequency spectral loss using a wavelet-based correction methodology (Nordbo and Katul, 2013). Average high-frequency loss factors for NO and $NO_2$ were found to be 1.014 and 1.015. As these corrections increased fluxes by only 1.4 - 1.5 %, they were not applied. A detailed overview of chemical and meteorological $NO_x$ flux losses can be found in Vaughan et al. (2016). As a final QA/QC filter, friction velocity (u*) was chosen as a metric of developed turbulence. A u* threshold of 0.15 m s$^{-1}$ was chosen in line with other urban EC studies (Langford et al., 2010; Squires et al., 2020).

### 2.2.4 Footprint model

To assess the spatial influence of each flux, we used a footprint model. The model calculates a spatial representative weighting matrix for each measurement along the flight track. In this study, we apply a model capable of assessing influence from





prevailing wind and crosswind (Metzger et al., 2012). The model uses a parameterised version of the Kljun (KL04) backwards Lagrangian model (Kljun et al., 2002, 2004), capable of calculating footprint estimates under stable and strongly convective conditions. Parameterisation was achieved using measurement height ($Z_m$), u*, standard deviations of vertical and horizontal wind speeds, the planetary boundary layer height ($Z_i$) and aerodynamic roughness length ($Z_0$). We used previously published

$Z_0$ values for the GLR, accounting for westerly and easterly wind influences, at 1 km$^2$ resolution (Drew et al., 2013). The model generates a weighting matrix across the same domain as the spatial dataset of interest, summing up to one and is centred on the aircraft's location. The footprint matrix can then be used to weight and cumulative sum the spatial dataset, giving a representative value along the flight leg. Fig. 4 shows the average calculated footprint across the campaign at 30, 60, and 90% influence contours. On average, the 90% influence distance ranged from 3 - 12 km.


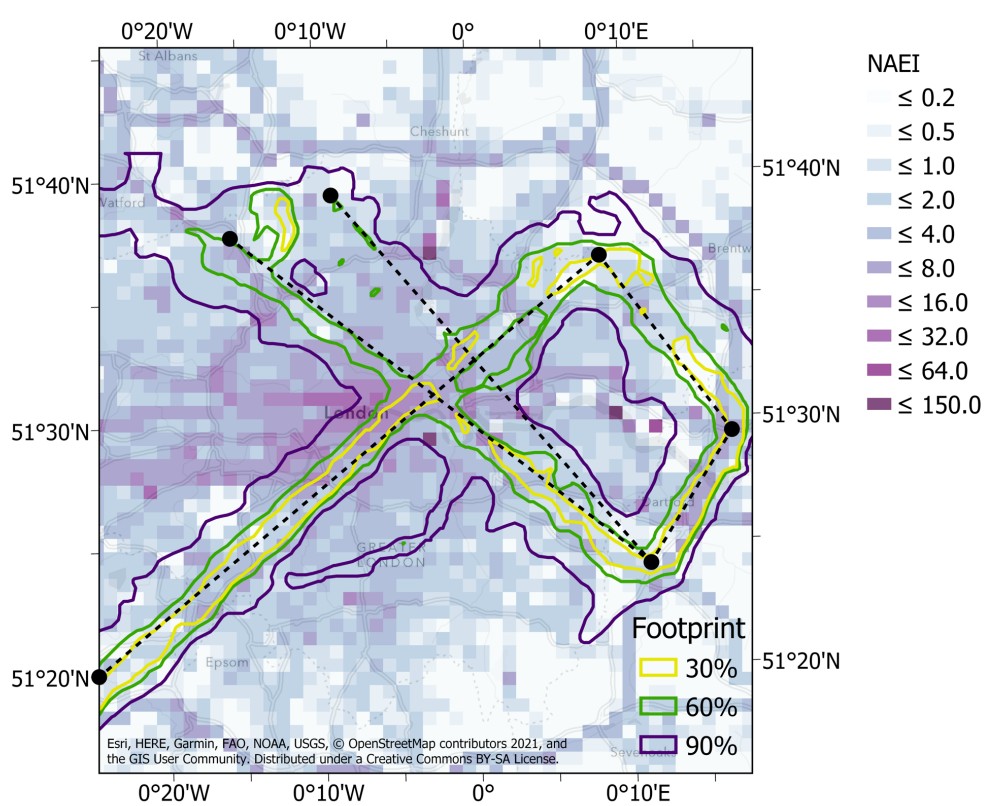

**Figure 4. Footprint climatology of all aircraft transects, indicated by the 30, 60, 90% contour lines of the cumulative surface influence superimposed over the 2014 NAEI for NO$_x$ emissions in Tons km$^{-2}$ yr$^{-1}$. Plotted in ArcGIS® (Esri, 2021).**



### 2.2.5 Boosted regression tree machine learning

Linking time-of-day measured fluxes at the aircraft transect height to the surface can be challenging and is driven mainly by their spatio-temporal variability. The application of an ERF, in contrast, can bridge this gap by building relationships between measured flux (spatial and temporal) and environmental drivers. We used boosted regression trees (BRT) (Elith et al., 2008; Metzger et al., 2013; Serafimovich et al., 2018) to calculate ERF relationships between measured airborne fluxes (spatial and temporal) and multiple environmental drivers. BRT is a non-parametric machine learning technique which combines regression trees and boosting to formulate ERF relationships (Serafimovich et al., 2018). BRT parameters were determined using the same strategy as Metzger et al., (2013), through the cross-validation procedure described in Elith et al., (2008). We found by using a learning rate of 0.1, tree complexity of 6, bag fraction of 0.75, absolute (Laplace) error structure and $3.7e^4$ trees overall, we were able to minimise the predicted deviance whilst achieving the optimum model fit. The BRT approach used an initial 500 trees, with 500 trees added at each step. The training dataset consisted of 1,751 airborne flux observations after QA/QC filtering.

## 3 Results and discussion

### 3.1 Airborne $NO_x$ fluxes

$NO_x$ fluxes were calculated during four flights, giving 11 complete transects across the GLR and 2884 individual 400 m flux averages. Measurements were made at a relatively constant altitude above the surface ($340 \pm 40$ m), corrected for both terrain elevation and building height. Building height data for the entire Greater London region was obtained from Digimap Ordnance Survey Web Map Service (Digimap) (Ordnance Survey, 2020). To account for changing boundary layer heights, we used hourly 0.25-degree estimates from the ERA5 fifth-generation ECMWF reanalysis for global climate data (Hersbach et al., 2018). Calculated depth of the boundary layer ($Z_m/Z_i$) ranged from 0.150 to 0.770, with a median $Z_m/Z_i$ of 0.255. Atmospheric stratification was found to be mostly unstable throughout the campaign, with a median Obukhov length (L) of -182 m and dimensionless Monin-Obukhov stability parameter ($Z_m/L$) of -1.98. Friction velocities ranged from 0.06 to 1.09 m s$^{-1}$, with an average of 0.56 m s$^{-1}$.

EC measurements are affected by random and systematic errors. Random error accounts for uncertainty due to insufficient averaging period, resulting in the inadequate sampling of primary contributing eddies (Lenschow et al., 1994; Mann and Lenschow, 1994). A detailed review of random error estimation approaches for EC can be found in Salesky et al. (2012). Systematic error accounts for under-sampling of the largest atmospheric scales responsible for turbulent flux (Lenschow et al., 1994; Mann and Lenschow, 1994). At a 400 m averaging interval, median random error for the NO flux was 126.6 % and 108.3 % for $NO_2$. The median systematic error for NO and $NO_2$ flux were 14.7 % and 14.3 %.



Each flux average only gives a temporally limited characterisation of a location's emission structure, leading to high uncertainties. By aggregating and averaging across multiple transects, temporal variability can be better accounted for. Fig. 5 shows mean 400 m latitude flux averages for each of the five transect types, with the shaded area the standard deviation of the calculated mean. Transect 1 follows an identical path to that of similar measurements made previously in 2013 and shows

comparable $NO_x$ fluxes (Vaughan et al., 2016). The highest observed fluxes (>20 mg m$^{-2}$ hr$^{-1}$) were measured over the London Borough of Southwark and the City of London. Both areas include major roads, national rail stations and densely packed high-rise buildings, giving profoundly heterogeneous emission sources of $NO_x$. Transects 2 & 3 (Fig. 5) ran perpendicular to transect 1, giving emission information over the South East and North West areas of Greater London. The emission structure of transect 2 shows similarities to that of transect 1, with fluxes in the central area above 10 mg m$^{-2}$ h$^{-1}$. Transect 3, in comparison, showed

50% lower emissions (5 mg m$^{-2}$ h$^{-1}$). This transect was over more suburban areas compared to transects 1 and 2. The final transects (4 and 5) ran over eastern parts of the GLR, extending out to the M25 Orbital Motorway and industrial infrastructure. The Dartford Crossing (A282) area showed elevated $NO_x$ emissions (>10 mg m$^{-2}$ h$^{-1}$). It was evident during most flights that this area was prone to congestion, suggesting vehicles as the primary source. The design capacity of the bridge is 135,000 vehicles per day, but vehicle flows now routinely exceed 160,000 per day.






**Figure 5. Ensemble 400 m NOₓ flux flight track averages across the OPFUE 2014 campaign. The top row (a, b & c) shows flight transects 1, 2 and 3 which ran over central areas of London such as City of London Borough. The bottom row (d & f) transects 4 and 5, ran over eastern regions of Greater London, home to industry and major road network. The shaded area represents the standard deviation of the ensemble mean. e) shows each individual track transect overlaid onto major road infrastructure, local boundaries, and rivers around the Greater London Region. Map was built using data from © OpenStreetMap contributors 2021. Distributed under a Creative Commons BY-SA License.**

## 3.2 Comparison to Emission Inventory

Measured fluxes are a powerful tool for evaluating bottom-up emission estimates, such as the NAEI. The NAEI is vital for assessing UK air quality, providing annual emissions estimates for a range of pollutants at 1 km² resolution for the UK region. Each pollutant has an individual bottom-up inventory, covering hundreds of different emissions categories, which, when summed together, give an annual national estimate. These sources include; road transport, domestic and industrial combustion,





rail, aviation, energy generation, waste, fossil fuel extraction and agricultural production. The NAEI's road transport sector is
based on emissions UK road traffic statistics and the COPERT (Calculation of Emissions from Road Transport) 4 emission
factor model, which is part of the European Monitoring and Evaluation Programme/European Economic Area (EMEP/EEA)
air pollutant emission inventory guidebook (Bush et al., 2008; EEA, 2013). For each airborne flux, a footprint matrix was
generated at the same spatial extent and resolution (1 km$^2$) as the NAEI. Each footprint equates to a value of one and weights
each grid cell of the NAEI individually. Once weighted, all cells are summarised, giving a spatially representative emission
estimate. We corrected for time-of-day emission variations by scaling each source sector for monthly, daily and hourly
influences. Once scaled, all sources are summed to produce a time-of-day estimate.

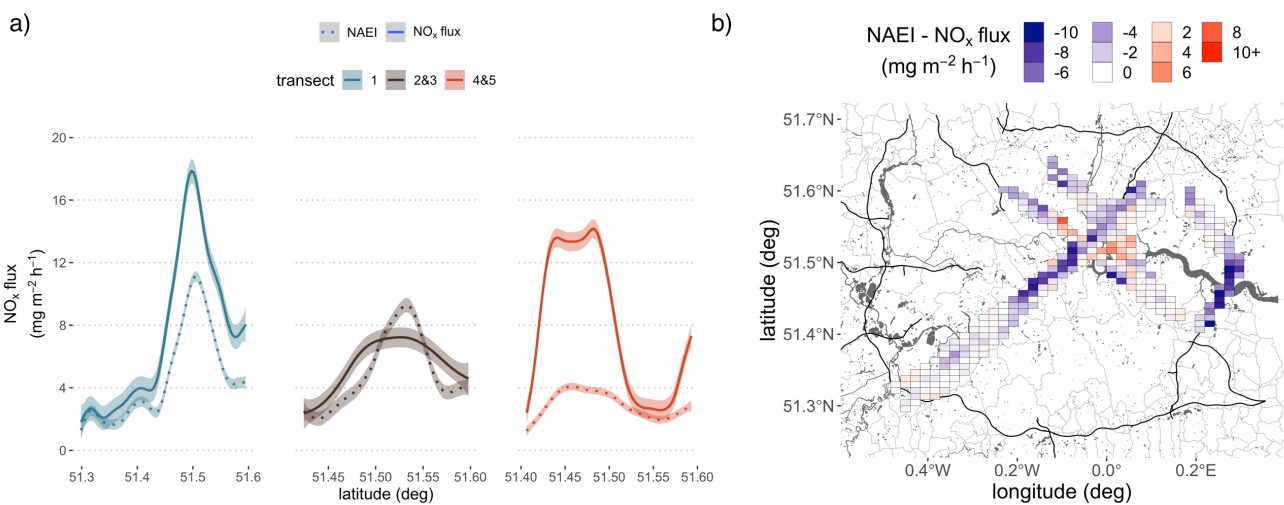

**Figure 6. a) Transect grouped measured NO$_x$ fluxes and NAEI emission estimates as a function of latitude. A generalised additive model (GAM) has been fitted to each transect grouping, using a k value of 10. b) Spatially median binned 1 km$^2$ difference between predicted NO$_x$ emissions (NAEI) and measured NO$_x$ fluxes, mapped onto major road infrastructure, local boundaries, and rivers around the Greater London Region. Map was built using data from © OpenStreetMap contributors 2021. Distributed under a Creative Commons BY-SA License.**


To compare measured fluxes against spatially representative NAEI estimates, each transects type was 1 km mean binned as a
function of latitude. Transects 2 and 3 were grouped to produce a perpendicular comparison to transect 1. Transects 4 and 5
were grouped to give a comparison in an area more representative of industrial/road transport-dominated emission sources.
Fig. 6a shows measured, and NAEI estimates as a function of latitude for each of the three groupings, with a generalised
additive model (GAM) fitted (Hastie and Tibshirani, 1990). Measured fluxes along transect one consistently showed higher
NO$_x$ emissions than estimated by the NAEI (mean of 1.5 times higher). The greatest divergence ratio between the measured
and inventory-estimate fluxes was 1.98, which is broadly consistent with previous studies (Lee et al., 2015). The divergence
for transect 1 was most substantial when a mix of different emission sources were encountered, such as other transport mediums



(rail and shipping) and, domestic and industrial combustion settings (see Table 3). Comparison for grouped transects 2 and 3 showed improved agreement to the inventory, with measured fluxes on average 1.21 times higher. The percentage contribution of emissions sources was similar to that of transect 1, with only a slightly lower average road transport contribution (63%). The stronger agreement between transects 2 and 3 suggests the high emissions observed during transect 1 are dependant on either a missing or under-represented source in the inventory. Grouped transects 4 and 5 also displayed a high degree of divergence from the inventory. On average, the ratio between measurement and inventory was 2.57, with a peak value of 4.45.

The primary sources for this area include a greater contribution from energy production and industrial combustion. Table 3 summarises for the three different groups, average NAEI sector contributions and the ratio between measurement and inventory.

| Transect | Road Transport | Other Transport | Domestic Combustion | Industrial Combustion | Energy Production | ratio |
|---|---|---|---|---|---|---|
| 1 | 63.89 % | 9.24 % | 21.71 % | 4.27 % | 0.82 % | $1.54 \pm 0.23$ |
| 2 & 3 | 62.75 % | 8.44 % | 22.2 % | 6.06 % | 0.42 % | $1.20 \pm 0.31$ |
| 4 & 5 | 70.09 % | 8.47 % | 11.1 % | 8.40 % | 1.90 % | $2.58 \pm 1.04$ |

**Table 3.** Transect grouping of NAEI predicted percentage emission contribution by five key sources and the mean ratio of measurement to NAEI estimate. These sources are; road transport, other transport such as rail and shipping, domestic combustion (combustion in commercial, institutional, residential and agriculture), industrial combustion (combustion in industry) and energy production (combustion in energy production and transformation).

Spatially, the disagreement between measurement and inventory is uneven, as shown by Fig. 6b, whether, for each 1 km along

the flight track, the median inventory minus measurement value has been calculated. South-western areas of the GLR agree better than the central and north-eastern areas. Greater under-estimation by the inventory compared with measurements was predominantly observed in regions of complex source distribution and where no single primary source dominated. The extent of disagreement highlights the challenges and consequent drawbacks of using the NAEI as a predictive tool for estimating $NO_x$ emissions or as a time-of-day diagnostic for measured $NO_x$ fluxes. Several vital processes may likely contribute to the observed

differences, in addition to $NO_x$ emissions being higher than in the NAEI. The first is inventory scaling from annual to time-of-day. As each source sector undergoes individual scaling, these factors play a significant role in predicting time-of-day influences. Currently, these factors lack spatial disaggregation and do not account for the unique temporal profiles present per area. In contrast to the NAEI, the London Atmospheric Emissions Inventory (LAEI) uses emissions data from individual vehicle classes, obtained by on-the-road 'remote-sensing', to constrain its predicted emissions from the road transport sector,

giving a more realistic comparison to "real life" emissions and hence to eddy-covariance measurements (Lee et al., 2015; Vaughan et al., 2016).



## 3.2 Spatio-temporal emissions

To overcome the limitation of using time-of-day representative NAEI estimates to explain measured fluxes, a more pragmatic approach was chosen. Using the outlined ERF methodology, we attempted to generate representative emission grids for each

flight transect. To train the BRT technique, $NO_x$ flux data was filtered to include 0.5 to 99.5% quantile values and positive fluxes only. We found excellent agreement between measured and ERF reproduced $NO_x$ fluxes, in the range of 0 - 37 mg m$^{-2}$ h$^{-1}$. The two datasets agreed on a 1:1 trend, with an $R^2$ coefficient of correlation of 0.99 and a residual standard deviation of 0.01.

Six environmental drivers were used in the ERF process to describe the spatio-temporal nature of the measured $NO_x$ fluxes. Fig. 7 shows the partial response functions calculated for each driver against difference from the mean flux and ranked in terms of percentage contribution to the flux distribution. Two different spatial datasets were used to account for the complex heterogeneity of the Greater London Region (Fig. 7a & c). Using the described footprint methodology, spatially representative surface $NO_x$ concentrations and building heights were calculated for each flux from the LAEI and Ordnance Survey datasets

(Greater London Authority, 2013; Ordnance Survey, 2020). Preliminary analyse using surface $NO_x$ concentration as the only spatial driver appeared to overweight suburban areas and underweight central areas of the GLR. The combination of the two datasets helps to reinforce the significant spatial differences between outer and inner London. To account for meteorological differences, $NO_x$ concentration at altitude (Fig. 7b), relative measurement height in the boundary layer ($Z_m/Z_i$) and potential temperature were chosen as ERF drivers (Fig. 7e & f). As shown in Fig. 7e, 90% of flight data occurs below a $Z_m/Z_i$ value of

0.4, with the function above 0.4 being mainly linear. Solar azimuth angle (Fig. 7d) was chosen to account for temporal variations in the measured flux. Flight data is well distributed across the solar azimuth angle domain from 100 to 260º, corresponding to 08:00-16:00 Greenwich Mean Time (GMT).

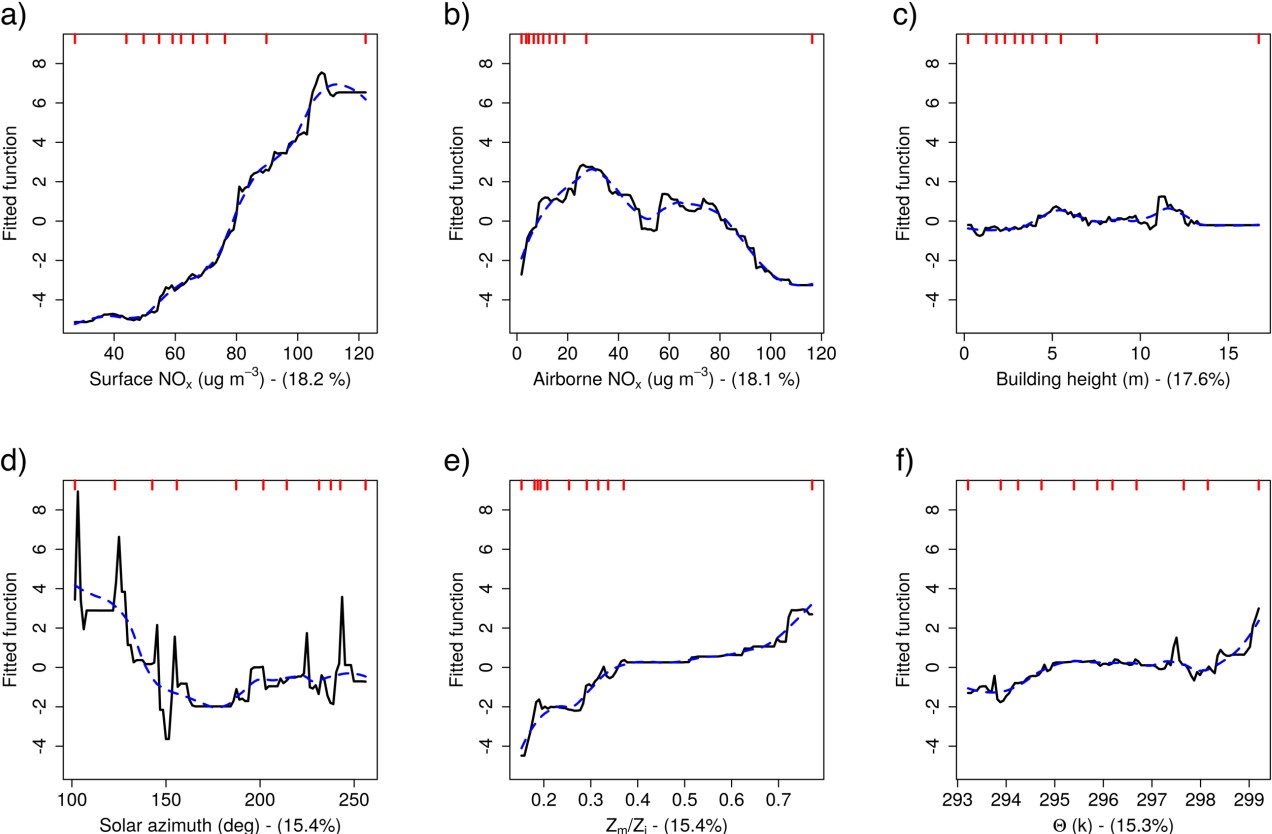

**Figure 7. Partial dependency plots for six environmental drivers, showing BRT-fitted ERFs for each driver as a function of flux dependency from the mean, and are ranked in terms of percentage contribution (%) towards accounting for NO$_x$ flux distribution. The red degree marks on the top x-axis show the data distribution from 0 to 100% in 10% bins.**

For each flight leg, surface-layer NO$_x$ fluxes were projected using median calculated statistics. Median values were chosen to account for the high heterogeneity across the length of a flight leg. Z$_m$/Z$_i$ values for each ERF flux projection were kept constant, to enable comparison between legs. Overall, 20 unique transects were projected onto an aggregated 400 m$^2$ LAEI grid, marrying to the spatial resolution of measured flux. Fig. 8 shows the median average of all ERF flux projections across the field campaign. Overall, ERF flux projection was possible across 98% of the GLR domain. Strong NO$_x$ emission rates are exhibited in central London with lower emissions in outer London. The standard deviation between individual flight transects is low, showing an of ± 2.45 mg m$^{-2}$ h$^{-1}$. The calculated relative standard deviation (RSD) shows a more complex picture, with predicted emissions in outer regions of London having a high RSD (>40 %) compared to central London (>35 %). Fig. 8C shows the calculated RSD across the GLR domain, suggesting central areas showed a more consistent emission profile during the campaign, highlighting the need for further refinement of how the ERF predicted emissions in outer areas of London. ERF did not extrapolate onto areas of much higher or lower surface NO$_x$ concentrations (shown as grey), which exceeded the ranges

observed in the training dataset. These areas included parts of the M25 orbital motorway, due to limited data airborne over the region and where footprints extended beyond the confines of the LAEI grid. Areas of central London are also left blank due to footprints not encountering surface concentrations above 122 ug/m³.

**Figure 8. Median (a), Standard Deviation (b) and Relative Standard Deviation (c) ERF flux projection from all individual flight legs at 400 m² resolution. Missing areas outside of the ERF training dataset are shown in grey.**

Diurnal variability was investigated during the campaign, by grouping flight data into hourly bins and using the median hourly statistics to drive each ERF flux projection. Again, $Z_m/Z_i$ was kept constant for all projections. Fig. 9 shows the average hourly

ERF flux projections, spanning an eight-hour period from 09:00 -16:00. All projections retain a strong heterogeneous profile. The most substantial emission rates were observed during 09:00 - 10:00 (Fig. 9a-b), aligning with the morning rush-hour. The emission rates rise across the GLR, in unison, until 10:00, when emissions stabilise into the afternoon period. Projected central London emissions during this period agree well with measured fluxes, whilst more suburban areas are potentially scaled too

high, suggesting further temporal refinement across the domain is required. The evening rush-hour, previously observed in

$NO_x$ emissions in London after 16:00 (Lee et al., 2015) is not captured in these predictions.

**Figure 9.** Hour-of-day ERF flux projections from 09:00 to 16:00. Grey colour highlight areas outside of the ERF training dataset. The strong presence of the morning rush-hour period is observed from 09:00 to 10:00 (a-b).

**4 Conclusions**

The assessment of $NO_x$ emissions in urban areas remains an important area for research, due to the critical impacts that high $NO_x$ concentrations have on local public health and the attainment of national trans-boundary emissions commitments. In this study, we used airborne measurements over the Greater London area to upscale airborne $NO_x$ flux observations to high-resolution emission projections across the region, via Environmental Response Function (ERF) physics-guided flux data

fusion. The work presented here presents a method which can quantify and spatially disaggregate $NO_x$ fluxes over challenging urban terrain and has the potential to be applied to other metropolitan areas worldwide.





Seven low altitude research flights were made over the Greater London region (GLR) in July 2014, performing multiple over-passes across the city. From these flights, 2715 individual $NO_x$ fluxes at 400 m spatial resolution were measured and processed in R using the eddy4R software. Measured $NO_x$ fluxes across the Greater London region exhibited high heterogeneity and substantial diurnal variability. Central areas of London showed the highest emission rates quantified during the campaign. Other high emission source areas included the M25 orbital motorway. The complexity of London's emission characteristics makes it challenging to pinpoint single emission sources definitively. In practice, multiple sources are likely to contribute to measured fluxes at the spatial scale used here, including road transport and residential, commercial and industrial combustion (mainly for space heating). To give a time-of-day reference, we compared measured fluxes to the UK's National Atmospheric Emissions Inventory, scaled to account for monthly, daily and hourly differences from the annual values. We found that for central areas of London, the inventory underestimated emissions by up to a factor of two, which is consistent with other published studies. Measured fluxes were consistently higher than inventory estimates across most of Greater London.

To overcome the limitations of comparing to the national inventory, we trained ERFs between measured spatial-temporal $NO_x$ fluxes and environmental drivers (meteorological and surface) to generate time-of-day emission surfaces. ERF successfully reproduced aircraft measured $NO_x$ fluxes, with a coefficient of determination ($R^2$) of 0.99. We used the calculated ERF relationships to project the $NO_x$ flux for the time of each flight transect across the GLR domain at 400 $m^2$ resolution. We were able to achieve a 98% spatial coverage and a highly heterogeneous emission surface. The overall variability between ERF flux projections was low, with an average relative standard deviation of 40%. All ERF flux projections showed high emissions emanating from central areas of London and the major road network. Hour of day projections highlighted a strong morning rush-hour, peaking at 10:00, and remaining elevated into the afternoon. Overall, the integration of high-resolution spatio-temporal fluxes with an ERF driven strategy has enabled the generation of spatial $NO_x$ emissions at high-resolution over Greater London.

This work demonstrates the power of airborne eddy-covariance based measurements of air pollutant fluxes as a tool for evaluating emission inventories or as a method of independently obtaining spatially disaggregated city-wide emission rates of pollutants. The method is applicable to other metropolitan areas or any other heterogeneous landscape. It should also help legislating authorities better understand air pollution sources and the effectiveness of control measures.



**Code availability**

The eddy4R v0.2.0 software framework used to generate eddy-covariance flux estimates is described in Metzger et al. (2017) and can be freely accessed at https://github.com/NEONScience/eddy4R. The eddy4R turbulence v0.0.16 software module for advanced airborne data processing described in Metzger et al. (2013) was accessed under Terms of Use for this study

(https://www.eol.ucar.edu/content/cheesehead-code-policy-appendix) and is available upon request.

**Data availability**

Any flux data presented here may be accessed by contacting the authors.

**Author contribution**

JDL, ACL, RMP, BD and CNH conceptualized the study and obtained funding. ARV, JDL, MDS, BD and CNH conducted the airborne field measurements. ARV, SM and DD analysed the eddy-covariance data and conducted the machine learning analysis. All authors reviewed and edited the paper.

**Competing interests**

The authors declare that they have no conflict of interest.

**Acknowledgements**

We thank the UK Natural Environment Research Council for financial support and the staff of the NERC's Airborne Research

and Survey Facility for their enthusiasm and skill in performing our multiple low-level flights across London. The National Ecological Observatory Network is a project sponsored by the National Science Foundation and managed under cooperative agreement by Battelle. This material is based upon work supported by the National Science Foundation (grant no. DBI-0752017). Any opinions, findings and conclusions or recommendations expressed in this material are those of the author and do not necessarily reflect the views of the National Science Foundation.






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
