# Peer review of "Spatially and temporally resolved measurements of $NO_x$ fluxes by airborne eddy-covariance over Greater London"

_Atmospheric Chemistry and Physics, 2021_

## Author Response (AR1)

**Authors' Response to Reviewer 1 (acp-2021-180)**

The authors' would like to thank the reviewer for their supportive comments and for taking the time to review the manuscript. Below is a breakdown of the reviewer comments in black with corresponding author responses in red. Line numbers given correspond to the revised manuscript.

**Specific Comments:**

1. L69: What is the typical aircraft speed?

The typical aircraft speed (science speed) was $74.5 \pm 10$ m s$^{-1}$, which has now been added to the revised manuscript. Line 73.

2. L71: What times of day did flights occur? This information should be added to Table 1 to give a better sense of diel sampling.

Table 1 has been updated to include additional information as to when each flight occurred, including the; date, weekday and specific hours for each.

3. Section 2.2: Much of the detail in this section could probably be moved to a supplement. Main points, including instrument accuracy and precision, should remain, but details on calibrations, dark counts, etc, read more like an instrument paper and distract from the main results.

Section 2.2 has been updated to keep a core description of the AQD instrument and its principles only. Discussion regarding calibrations, dark count assessment, and the flow schematics (Fig. 2) has been moved to the supplementary, listed as section 1.1. Section 2.2 should now read more clearly and focus more on instrument uncertainties than NO$_x$ analysers internal workings.

4. L114: Is the 3.3% correction referenced here to generate dry mixing ratios from wet? Or is this the correction due to quenching of the chemiluminescence reaction from water vapor? If the former, although the correction is small, the authors should still note that unless using dry mixing ratios to generate NOx fluxes, they may need to apply a Webb density correction to account for heat and water vapor. No mention of a density correction is made later on in the text.

The 3.3% correction has been removed from the manuscript, with a more detailed discussion now appear in Section 1.1 of the supplementary material, discussing the water vapour addition to the Fast-AQD-NOx as a stability method in order to negative changes in chemiluminescence quenching. Section 3.1 (lines 287-291) now contains a detailed assessment of WPL corrections for the measured NO$_x$ fluxes, using the method outlined by Hartmann et al., 2018. Fig. S5a in the supplementary shows the effect of using wet vs dry mole mixing ratios for calculating NO$_x$ fluxes. Correcting for WPL increased measured NO$_x$ flux on average by 1.35%.

5. L133-134: Detection limits of 49 and 78 pptv for NO and NO2 seem unrealistically low for a 9 Hz integration time. Another reference from the paper using the same instrument (Lee et al, 2009) quotes a 2-sigma LOD for NO of 36 pptv at 1Hz. If this is true, adding the Allan-Werle plot to the supplement would be useful.

The 2$\sigma$ precision has been reanalysed using in-flight zero data from across the campaign. Fig. S2 in the supplementary shows for each flight the density distribution of the zero counts. The updated 2$\sigma$ was found to be 153 and 249 ppt for NO and NO$_2$ (for a 9 Hz integration time). It should be noted that the PMT temperature of the detector is significantly lower ($<$-60 oC) than that of the system reported in Lee et al. (2009) (-25 ºC), which will give better signal stability.

6. L145: This is not a strictly correct error estimation. The individual uncertainties should be propagated through all the equations used to calculate NO/NO2 mixing ratio.

The manuscripts original words was not clear, with the discussed overall uncertainties being the propagation method. Using the recalculated $2\sigma$ precision, an overall uncertainty for 1 ppb of NO and $NO_2$ is now listed in Section 2.2 (Line 129).

7. L149-150: Which species were measured by the PTR-MS and Picarro? Were they used in this analysis at all?

Data collected by the PTR-MS has been discussed already elsewhere, with the papers discussing various anthropogenic and biogenic non-methane VOCs (Shaw et al., 2015; Vaughan et al., 2017). Data collected by the Picarro and PTR-MS is not used in this study. The manuscripts text has been updated to clarify this (lines 137-138).

8. Section 2.2.3: Was there any treatment or consideration of a vertical flux divergence? This is an important point that the authors should address.

Fig S4b in the supplementary material shows the effect correcting for vertical flux divergence could have on calculated $NO_x$ fluxes (up to a 50% increase) using the Sorbjan (2006) method. As the boundary layer estimates in this study are from the ER5 reanalysis dataset, there is potential for large uncertainties compared to in-situ LiDAR measurements. There is also high uncertainty as to the effect flux storage has in highly complex urban terrains, such as from street canyons. Future studies are needed to assess these processes in greater detail. Fluxes reported in this study are classified as conservative, with any vertical divergence processes having the potential to increase measured fluxes and the discrepancy between measurement/inventory.

9. L225-226: "whereas time-frequency EC gives a flux measurement every 400 m along the transect using a 4000 m moving window…" How is the 4000 m moving window applied to the 400 m CWT fluxes? Are the measurements overlapping? This wording is unclear.

Yes, the 400m window will lead to flux estimates will overlap. The manuscript text has been updated to clarify this.

L245: One of the advantages of the CWT method is that by decomposing the signal into the time domains, the strict criteria for stationarity is not necessary.

Fluxes were flagged at the 100% stationary mark, but most flight legs that failed the stationary criteria had already failed other filtering criteria as discussed in section 2.2.3. Therefore, we don't believe that this QA/QC method will lead to incorrection data filtering.

10. L250: It's not explicitly stated, but was data with u*<0.15 m/s discarded?

The manuscript text has been clarified to show that data below 0.15 was filtered out. A 0.15 threshold was used to mirror other urban studies in London as a developed turbulence metric.

11. Section 3.1: Some additional quality metrics for the NOx fluxes would strengthen the results. Examples include a lag-covariance plot, a CWT cross-scalogram, wind and scalar power spectra and co-spectra. Such figures could be in the supplement but would build confidence in the application of the eddy covariance technique to NOx fluxes and would give a visual idea of signal-to-noise.

We have added additional flux quality metrics to the supplementary material (Section 1.2). These include; lag-covariance plots, a CWT cross-scalogram and the average cospectra for $NO_x$ and heat flux. These metrics support the strength of the present $NO_x$ fluxes as a good quality dataset.

12. L299-300: The point-by-point errors are significant. How do errors reduce when averaging? Figure 5 shows the standard deviation in shading, but it would be helpful to get a sense of the error when averaging all transects of a given type together. These uncertainties should be reported, even if not displayed in the figure.

Figure 5 has been updated to show the average flux random error divided by the square root of the number of sample points which went into each mean (shaded area). This gives a better visualisation of the overall uncertainty of the flux averages. By averaging multiple transects together, the temporal variability between legs is reduced, provide a more accurate spatial picture. The individual uncertainties of a single flux measured are also discussed in Section 3.1 (lines 245 – 252).

13. L334-335: In general, not much explanation is given in the manuscript about when sampling occurred. At what times are the measurements acquired? How is averaging performed across transects sampled at multiple times of day? Doesn't this dampen any diel variability? How is the timing compared to the emissions inventory? Are you comparing emissions for each transect time independently or to a mean? It would be helpful to add some clarity on these points in the text.

Section 3.1 has been updated to include an additional figure (now called Fig. 4), discussing the temporal variability of measurement $NO_x$ flux in three distinct areas during the campaign (lines 270 – 284).

Comparison between measured fluxes and annual emission inventories was achieved by calculating an individual inventory estimate for each snap sector (different emission sources) using the outlined footprint methodology in section 2.2.4. Each snap sector estimate is then weighted using source-specific scaling factors that account for; monthly, daily and hourly variations. These scaled estimates are then summed up to give a time-of-day inventory estimate, accounting for the location and time-of-day that the flux measurement was made.

14. Figure 6: What does the shading represent? Is it the uncertainty of the measured/inventory fluxes? Also, please elaborate further on the GAM. There is not much description in the text. Are the NAEI estimates here generated from the flux footprint?

The manuscript text has been updated to clarify that time-of-day scaled NAEI estimates are footprint calculated (lines 352-355). The GAM models fit to each dataset have also been further discussed, with the shaded areas representing the 95% confidence interval of the GAM fit.

15. L410: The description of Figure 8 is unclear, particularly the phrase "median average". Please elaborate on what is being depicted.

The figure caption has been updated to clarify which median average is being shown. The median average is the average of all individual flight transect projections using the ERF approach.

16. Figure 8: Should specify that these are NOx emission rates. What is the uncertainty in the flux projection? It would be helpful to see a map depicting the associated uncertainties. The standard deviation only indicates the variability between different legs but does not help quantify how any uncertainty in the measurement propagates into the ERF model.

Figure 8 has been updated to specify that $NO_x$ emission rates are being shown. Using the method outlined in Metzger et al., (2013), the variability in the BRT model performance was assessed by individually omitting one flight leg at a time and using the incomplete model to predict the omitted flight leg. The median difference between complete and incomplete model prediction was 13.7%, which is comparable to the model differences observed for sensible and latent heat flux (11-18%), using the same technique.

17. L431: It would be helpful to show a figure depicting the comparisons between ERF-reproduced and measured fluxes in the supplement to get a visual idea of how robust the ERF technique is.

Figure S7 has been added to the supplementary, showing each predicted flight leg emission map, with measured $NO_x$ fluxes overlaid. The majority of flight leg projects successfully scaled Central London emissions comparably to that of measured fluxes. The projects also successfully captured key features in the flux observation, such as major road networks and densely populated areas.

**Technical Corrections:**

1. Figure 1: Lat/lon coordinates for each transect do not align with those listed in Table 1.

Both Table 1 and Figure 1 have been updated to use the same decimal degree coordinate system.

2. Table 1: Add typical transect altitude or range of altitudes. Add time of day each transect was sampled.

Table 1 has been updated to include altitude information for each flight.

3. Table 2 is incorrectly labeled as Table 1.

The naming error for table 2 has been corrected.

4. Table 3: Label column "ratio" with something more descriptive.

This has been clarified in the manuscript text. This now reads the ratio (flux/naei).

5. Figure 6: Describe what shading indicates in the caption.

Shaded is now discussed in the manuscript text (lines 339-3) and figure caption, representing the 95% confidence interval of the GAM fit.

6. Figure 7: Identify blue and black curves in the caption.

Blue and black lines have now been discussed in the figure caption.

7. Figures 8 & 9: Specify NOx with units or in captions

Correct units have added to figure captions.

**Authors' Response to Reviewer 2 (acp-2021-180)**

The authors' would like to thank the reviewer for their supportive comments and for taking the time to review the manuscript. Below is a breakdown of the reviewer comments in black with corresponding author responses in red. Line numbers given correspond to the revised manuscript.

**Specific Comments:**

1. Introduction: it will be more valuable to summary more about the significance and the current progress of NOx flux measurement in complex terrain and then the advantage of airborne eddy-covariance approach should be emphasized.

The introduction section of the manuscript has been updated to have a short discussion as to the current status of $NO_x$ flux measurements, their focus and the next steps, which this manuscript presents in linking complex $NO_x$ emissions to heterogeneous urban topography (lines 60-66).

2. Methods: the general description on the AQD instrumentation and the methodology of eddy covariance with environmental response functions (ERF) can be shortened as it is well established instruments and software packages. In contrast, more information on the use of AQD and the specific improvement on the ERF as well as the aircraft cruises (flight time, heights, speed, etc.) shall be added.

Section 2.2 has been updated to keep a core description of the AQD instrument and its principles only. Discussion regarding calibrations, dark count assessment, and the flow schematics (Fig. 2) has been moved to the supplementary, listed as section 1.1. Section 2.2 should now read more clearly and focus more on instrument uncertainties than $NO_x$ analysers internal workings. Table 1 has been updated to include more information as to when each research flight occurred. The ERF model is discussed in detail in section 3.2 of the manuscript.

3. Methods: the specification of 9 Hz for the AQD instrument as the data frequency (time resolution) is not appropriate. The 9 Hz is the data acquisition rate but the input values of edd4R requires the time resolution. Sum up all the residence time of the air samples in the AQD shall be roughly the response time and the time resolution will be even longer than that.

The resonance time of the instrument's $NO_2$ converter is 0.11 s, and so this is taken as the appropriate sampling rate (9 Hz). This is now stated in the manuscript text.

4. Results: the authors may show and analyze their original observations of NO and NO2 before the results of EC deduced NOx flux. There are plenty of information from the directly observed values of both NO and NO2. For example, the NO2/NO ratio will give you some estimates of the observed air mass ages. To show the directly observed values are also a check of the data quality for the subsequent EC analysis. The flight legs may cross different terrains. The observed NO and NO2 may be different for traffic (surface) and industrial (point) emissions.

The manuscript's focus is on calculating $NO_x$ emission rates and assessing the Spatio-temporal variability in relation to London's heterogeneous emission surface. Considering calculated fluxes are compared to the NAEI, which already contains multiple emission source information, we don't feel adding concentration data will benefit the emission assessment being described in this study.

5. Results: Are there nighttime flights or twilight flights? The stated minimum flight height is around 300-400 meter which is limited in the range of nocturnal residual layer and the NOx emissions may not be measured precisely. The needs additional explanations.

Due to significant air traffic control restrictions over London, no nighttime flying was possible. Therefore, the flying hours as shown in the updated Table 1, ranged from 08:00 to 16:00 only.

6. Results: The solar azimuth angle seems to be a dominant factor for the NOx emission rates which means a strong diurnal pattern of the varied emissions sources. Some in-depth discussions on this feature may be added in the paper.

Section 3.1 has been updated to include an additional figure (now called Fig. 4), discussing the temporal variability of measurement $NO_x$ flux in three distinct areas during the campaign (lines 270 – 286). The strong-temporal variability fits with the solar azimuth angle be a dominant factor in the measured $NO_x$ fluxes.

7. Results: in previous publications by the same group, tower platform has also been used by the same group in GLR to deliver the NOx emission flux. Will there be a better idea to pin point the key environment factors of NOx emissions?

Measurements at the BT Tower in London did not occur during the flight period discussed, unfortunately. Future work within the group may want to look into coupling both types of measurements and applying ERF analysis from the tower to extract key emission factors as suggested, but at this moment in time, this is out of scope for this study.

**Technical Corrections:**

1. On line 17, "the Greater London region" is mentioned here for the first time, the abbreviation "GLR" should follow.

This has been updated to use GLR abbreviation after line 17.

2. On line 35 the second page, extra spaces appear in brackets.

This has been corrected.

3. On line 80 Table 1, the first and last line of the table needs to be upper/underlined. Besides, all the other tables in the main text should be modified to an identical format.

All tables have now been updated to use this formatting.

4. On line 113, please add more detailed descriptions on how the 3.3 % correction was calculated.

This statement has now been clarified by discussing the effect of WPL corrections in Section 3.1. Figure S4a in the supplementary shows the effect of correcting from wet mole to dry mole NOx and the effect it has on the calculated flux, which is less than 1.5%.

5. On line 122, please unify the tense of the sentences of the experiments. This sentence uses the present tense, while the previous sentence uses the past tense.

This information now appears in section 1.1 of the supplementary and has been updated to use the correct tense.

6. On line 128, please provide specifications about the UV pen ray lamp.

Section 1.1 supplementary now contains information about the UV pen ray lamp with a central wavelength of 254 nm.

7. On line 235, shall be table 2.

This has been corrected to read table 2.

**List of Manuscript Changes**

1. A new paragraph has been inserted at line 60, outlining current NOx emission studies and the focus of the manuscript.

2. Text has been inserted at line 69, outlining the science speed of the aircraft.

3. Text inserted at line 79, outlining which flights are discussed in the manuscript.

4. Table 1 has been reformatted to include addition about flight times and altitudes.

5. Section 2.2 has been reduced. Lines 105 to 128 have been moved to the supplementary material.

6. Figure 2 has been moved to the supplementary material.

7. Lines 144 – 146 have been changes to clarify the total uncertainty of measured NOx mixing ratios.

8. Additional text has been added to line 154, outlining which instrument techniques are discussed in the manuscript.

9. Lines 167 – 206 have been moved to the supplementary material.

10. Heading 2.2.1 now reads 2.3.1. Line 166.

11. Figure 3 is now labelled as figure 2.

12. Additional text added to line 226, discussing the choise of the maximum wavelet scale.

13. Text has been inserted at line 236, discussing the use of lag covariance plots as a QA/QC metric.

14. Text inserted at line 246, discussing u* filtering as a QA/QC metric.

15. Figure 4 is now labelled figure 3. Updated within manuscript text.

16. Added text at line 291 discussing individual flux uncertainties and chemical losses.

17. A new paragraph has been inserted at line 295, discussing WPL and vertical flux divergence.

18. A new paragraph has been inserted at line 296, discussing the temporal component of measured fluxes.

19. New Figure (Figure 4) inserted at line 297.

20. Inserted addition error discussion at line 298-299.

21. Updated Figure 5 caption.

22. Clarified text discussing footprint calculated NAEI emission estimates. Lines 327 – 331.

23. Updated Figure 6 caption.

24. Added information relating to GAM model using in Figure 6. Line 345.

25. Updated Table 3 header and caption information.

26. Updated text on lines 381 – 382, discussing BRT model performance.

27. Updated Figure 7 caption.

28. Updated Figure 8 caption.

29. Inserted addition paragraph at line 423, discussing the BRT models predictive uncertainty.